# Marine biofilms: cyanobacteria factories for the global oceans

Cheng Zhong,[1,2] Shun Yamanouchi,[3] Yingdong Li,[1,2] Jiawei Chen,[1,2] Tong Wei,[1,2] Ruojun Wang,[1,2] Kun Zhou,[1,2] Aifang Cheng,[1,2] Weiduo Hao,[4] Hongbin Liu,[1,2] Kurt O. Konhauser,[4] Wataru Iwasaki,[3,5] Pei-Yuan Qian[1,2]

**ABSTRACT** Marine biofilms were newly revealed as a giant microbial diversity pool for global oceans. However, the cyanobacterial diversity in marine biofilms within the upper seawater column and its ecological and evolutionary implications remains undetermined. Here, we reconstructed a full picture of modern marine cyanobacteria habitats by re-analyzing 9.3 terabyte metagenomic data sets and 2,648 metagenome-assembled genomes (MAGs). The abundances of cyanobacteria lineages exclusively detected in marine biofilms were up to ninefold higher than those in seawater at similar sample size. Analyses revealed that cyanobacteria in marine biofilms are specialists with strong geographical and environmental constraints on their genome and functional adaption, which is in stark contrast to the generalistic features of seawater-derived cyanobacteria. Molecular dating suggests that the important diversifications in biofilm-forming cyanobacteria appear to coincide with the Great Oxidation Event (GOE), "boring billion" middle Proterozoic, and the Neoproterozoic Oxidation Event (NOE). These new insights suggest that marine biofilms are large and important cyanobacterial factories for the global oceans.

**IMPORTANCE** Cyanobacteria, highly diverse microbial organisms, play a crucial role in Earth's oxygenation and biogeochemical cycling. However, their connection to these processes remains unclear, partly due to incomplete surveys of oceanic niches. Our study uncovered significant cyanobacterial diversity in marine biofilms, showing distinct niche differentiation compared to seawater counterparts. These patterns reflect three key stages of marine cyanobacterial diversification, coinciding with major geological events in the Earth's history.

**KEYWORDS** marine biofilms, cyanobacteria, metagenomics, biodiversity, evolution

Cyanobacterial genomes and their ecology and evolutions are emerging proxies to promote the understanding of geological events such as Earth's oxygenation (1). Since the Archean, cyanobacteria are likely among the most diverse and abundant microbes in the marine environment, colonizing vast swaths of the marine photic zone to sediments and other solid substrata in estuarine and coastal waters (1, 2). A previous experimental study of oxygenation from multiple benthic environments has suggested that benthic biomass played a crucial role in the initial rise of planetary oxygenation during the Archean, contributing significantly to the Great Oxidation Event (3). This finding was supported by phylogenetic and molecular clock analyses of cyanobacterial genomes available in the public databases, and the genome analyses further suggested that the emergence of planktonic cyanobacteria was associated with the Neoproterozoic Oxidation Event (4–6). Changes in oceanic chemistry and biosphere are plausible reasons for the dramatic divergence of the benthic and planktonic cyanobacteria (7, 8). However, the understanding of the full extent of how both benthic and planktonic cyanobacteria have interacted with Earth's evolutionary trajectory over a long geological period, such as the Proterozoic "boring billion" (1.8–0.8 Ga) remains limited.

Address correspondence to Pei-Yuan Qian, boqianpy@ust.hk.

The authors declare no competing interests.

See the funding table on p. 14.

With the rapid increase in available genomic and metagenomic data, there is a significant opportunity to uncover new insights into cyanobacterial eco-evolutionary patterns and their interactions with Earth's atmospheric and oceanic changes (9–11). For example, *Tara Oceans* project analyzed 7.2 terabyte of metagenomic data to illuminate microbial diversity for global oceans (12).Following this, surface-associated microbes have increasingly been recognized for their profound historical and ongoing impact on the Earth's environment (13, 14). Within the surface-associated microbes' categories, more than 7,300 "species" identified in the marine biofilms are undetected in seawater, which implies marine biofilms are previously underappreciated niches of novel microbial species and functional potential (15). Marine biofilm microbes offer numerous ecological and evolutionary advantages, such as environmental protection, enhanced nutrient access, and increased interaction opportunities with other organisms, making them important forces for Earth's evolution (16). Despite these advancements, comprehensive analyses of cyanobacterial genomes within these metagenomic data sets are lacking.

Here, we reprocessed a previously established terabyte-scale metagenomic data set, comprising 101 marine biofilm and 91 seawater samples across Earth's oceans. Specifically, we quantified marine cyanobacteria diversity and their ecological characteristics to reveal if the marine biofilm serves as an important source of the taxonomic and functional diversity of marine cyanobacteria. Insights into phylogenetic relationships and the possible origin of the marine biofilm cyanobacteria were also obtained by reconstructing molecular-dated trees. Ultimately, we aim to use metagenomic data to elucidate detailed cyanobacterial divergences, their connections to key geological events, and highlight previously underexplored periods linking cyanobacterial evolution with Earth's geological history.

## MATERIALS AND METHODS

### Data source

The metagenomic data of the 101 biofilm and 91 seawater samples were described in previous studies (15, 17). Briefly, samples were originally collected from 10 marine regions (Fig. 1). For marine biofilm samples, three samples from Sapelo Island, North Atlantic Ocean; 12 samples from Red Sea; 70 samples from Hong Kong water, South China Sea; one sample from Yung Shu O Bay, South China Sea; and five samples from Zhuhai Xiangzhou Bay, South China Sea; five samples from South China Sea; and six samples from East China Sea. A total of 101 samples were developed from seven substrata, including Petri dish (70 samples), zinc panel (12 samples), aluminium panel (two samples), poly(ether-ether-ketone) panel (six samples), titanium panel (two samples), stainless steel panel (two samples), and rock panel (seven samples). The duration of the biofilm development has a range from 3 days to >30 days (rock panel). Besides, 91 of the 267 total seawater samples available in *Tara* Oceans, which were sampled nearby the marine biofilms, were also collected for parallel analysis. As reported previously, the sequencing depth of these biofilm and seawater samples had sufficient coverage for a thorough survey of entire microbial communities and allowed the comparison of these samples (15).

### 16S rRNA Illumina tag analysis

#### Taxonomic profiling

16S rRNA Illumina tags (miTags) were extracted from the unassembled metagenomic data, and the taxonomic classification adopted herein followed a previous study (15). The 16S rRNA miTags were mapped to a database that integrates GreenGenes with RDP and SILVA. OTUs with 97% similarity or above were selected. The 16S rRNA miTags predicted to the cyanobacteria phylum were selected, and the OTUs classified to chloroplasts were further trimmed off for all subsequent analyses owing to their eukaryotic origin. A Venn diagram implemented in R v.4.0.5 was used to assess the distribution of full OTUs across

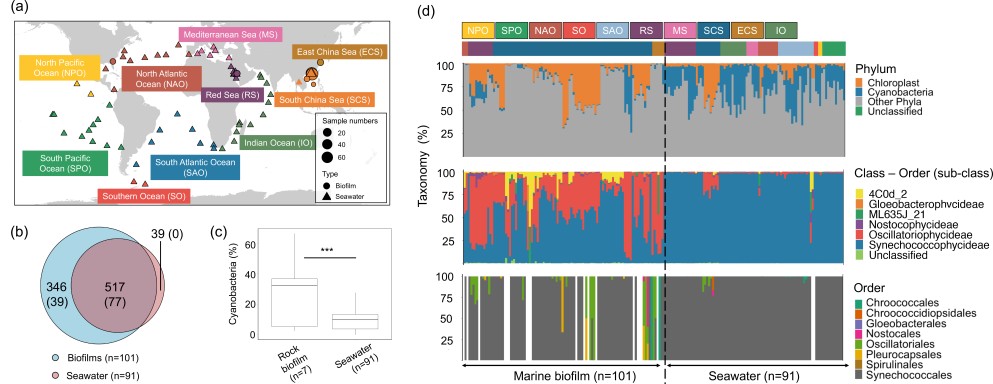

**FIG 1** Cyanobacteria species across marine biofilm and seawater samples. (a) Sampling locations of the metagenomes. (b) Venn diagram showing the distribution of cyanobacterial 16S rRNA miTag OTUs and full-length 16S rRNA gene units (values presented inside parenthesis) across 101 biofilm and 91 epipelagic seawater of the total seawater samples at approximately equal size conditions. (c) Relative abundance of cyanobacteria on rock surfaces and seawater samples. (d) Phylum- and subclass-level (16S rRNA miTags) and class-level (protein marker genes) taxonomic profiling showing substantial differences in cyanobacterium community compositions between the biofilm and seawater samples. The color and text codes in the upper part of the taxonomic profiles refer to the oceans.

the 101 marine biofilm and 91 epipelagic seawater samples. The sequences of 101 marine biofilm and 91 epipelagic seawater samples were normalized to relative abundance for taxonomic profiling at the phylum, subclass, and genus levels. Metagenomic reads were also mapped to 40 universal single-copy marker genes to cross-validate the 16S rRNA miTag-based taxonomic profiling using mOTU v. 3.0.0.

### Diversity, network, Mantel, and occupancy analyses

The metagenomic data of 101 marine biofilm and 91 epipelagic seawater samples were normalized to 10,000 sequences per sample for subsequent biogeographical analyses. Observed OTU, Chao1 richness, and ACE richness indices were used to compare community richness, whereas Shannon and inverse Simpson indices were utilized to compare community diversity. Jaccard distance was employed to compare OTU similarity. These diversity indicators were analyzed using phyloseq v.1.22.3 (18). The Bray–Curtis-based PCoA ordination with 95% confidence intervals was generated for dissimilarity analyses and pairwise comparison of marine biofilm and seawater cyanobacteria communities. The taxonomy-based PCoA ordination (Bray–Curtis distance) was generated to search for microbial species separating marine biofilm and seawater samples.

Network analyses of the OTUs associated with cyanobacteria from the normalized 16S rRNA miTags for the 101 biofilm and 91 epipelagic seawater samples were conducted using igraph (19) implemented in phyloseq. The Bray–Curtis distance and the maximum distance of 0.9 were used to build a network map. The Fruchterman–Reingold layout was utilized for network visualization. The objective distance and overall network topology, including average degree, closeness centrality, between centrality and eigenvector centrality, and between the biofilm and seawater samples, were compared.

Changes in the OTUs' co-presence numbers were analyzed across different sizes of the marine biofilm and seawater samples. With regard to changes in the biofilm and seawater samples, the linear regressions were used to assess their tendency from being generalists to specialists. The results of these analyses were confirmed by analyzing the non-normalized 16S rRNA miTags data set. Changes in the mean abundance of the OTUs were constructed as a function of their co-presence in the increased samples. The cutoff values for generalists and specialists were obtained from previous studies (20, 21). Cyanobacteria species were defined as generalists when they were present in >75% of the total sample size, whereas they were defined as specialists when they were present

in <25% of the total sample size with a relative abundance of >100 in log 10 scales. This applied classification aimed to distinguish the distribution of the generalists and specialists in the biofilm and seawater samples.

Partial Mantel correlations between the compositional data and the geographic distance (9,999 permutations) of the metagenomic data were computed using functions implemented in R. The Bray–Curtis distance and the Haversine distance were used for the compositional data and the geographic data, respectively. Then, seawater subsamples with coordinates similar to the biofilm samples, including samples falling in the intervals of the coordinates 0–2,500,000 km, 7,000,000–8,000,000 km, and 11,000,000–15,000,000 km, were collected to make the marine biofilm and seawater data sets more comparable to each other. Linear correlations were built for each data set. A Spearman-based matrix of correlation was also constructed across each phylum-level taxa for both samples by using the Hmisc package implemented in R (22).

## Analysis of full-length 16S rRNA gene sequences

### Distribution of full-length 16S rRNA gene units

The full-length sequences of 101 biofilms and 91 seawater samples were obtained using PhyloFlash v.3.0, and replicated sequences (100% similarity) were removed using CD-HIT (23). The method was adopted to build a Venn diagram as that for OTUs.

### 16S rRNA gene phylogenetic analysis

As for full-length 16S rRNA gene trees (posterior probability >80%), sequences were assembled from shotgun metagenomic reads of the 101 biofilm and 91 seawater samples and then clustered with 97% sequence identity to obtain 62 and 13 cyanobacterial representative sequences, respectively. The software used in this process was PhyloFlash v.3.0 with default settings, which was integrated with various types of software (e.g., SPAdes) and assigned the taxonomy on the basis of the modified versions of the SILVA SSU database (24). The sequences of the 16S rRNA genes associated with cyanobacteria, except for chloroplast, were selected for further analysis. This method minimized bias produced during the binning process of each metagenome sample, which supplemented useful results and additional validation for subsequent MAG-based analyses.

### Alignment, Bayesian tree inference, divergence time estimation, and calibration

Briefly, Bayesian analysis of 16S rRNA genes was performed in three steps: first, inference of the tree topology, dating using the correction points indicated by a previous study, and finally estimation of the diversification rate along branches. We reconstructed a non-clock molecular phylogenetic tree to infer cyanobacterial 16S rRNA phylogenetic relationships (i.e., tree topology). The sequence data set consisted of 42 metagenome-derived cyanobacterial sequences clustered by 97% identity, 56 cyanobacterial sequences from NCBI GenBank, and three outgroup sequences (4). Sequences were aligned with MAFFT v.7.480 (25, 26) and subjected to the tree inference without gap trimming. Bayesian tree inference was performed with MrBayes v.3.2.7a (27) under the GTR + I + G4 model to obtain a 50%-majority rule consensus tree. Detailed methods to build the phylogenetic tree are presented in the supplemental material.

The divergence time of cyanobacteria was then estimated under a relaxed molecular clock. After removing three outgroup sequences from the data set, we again generated a multiple sequence alignment. Bayesian molecular clock analysis was conducted using BEAST v.2.6.3 (28). The GTR + I + G4 model was specified for the substitution model, and the clock rates were assumed to follow the uncorrelated log-normal distribution. Several constraints and calibration points were also imposed according to "Analysis 7" described in the previous study (4) after confirming that the phylogenetic tree obtained from the non-clock analysis was consistent with their trees for major subclades. MCMC runs were performed to obtain the maximum clade credibility tree. Note that we performed

another MCMC run in advance under a more moderate prior distribution to prepare a starting tree that satisfies their strict assumptions. For more details, see the supplemental material.

To improve the taxonomic resolution of seawater-derived cyanobacteria, we attempted the same analysis on a data set with metagenomic sequences clustered by 99% identity. However, we could not reconstruct the non-clock tree on this data set because MrBayes did not produce any properly converged MCMC run. The 97% identity sequence clusters and the 99% identity sequence clusters were compared to confirm that there was no discrepancy that would affect the calibration points or monophyletic constraints. Thus, the non-clock analysis was skipped, and BEAST clock analysis was directly performed to obtain a dated tree with a high resolution successfully.

### Estimation of linage-specific diversification rates

The presence (or absence) of any phylogenetic disparity among cyanobacterial clades was tested by estimating lineage diversification rates via an established method (29), which modeled speciation, extinction, and change in rate categories as possible events along tree branches (the birth–death–shift process). Note that methods for modeling state-dependent diversification rates, such as the binary-state speciation and extinction (BiSSE) model, were not applicable to our data set because it contained too few state transitions that could be used to estimate parameters accurately.

The metagenomic samples were assumed to have uniformly sampled all extant cyanobacteria in the hydrosphere. On the basis of this assumption, the phylogenetic tree was pruned to include only metagenome-derived 16S rRNA clusters with 99% identity. With this phylogenetic tree as the input, Bayesian model inference was performed using RevBayes v.1.0.13. by following the tutorial (https://revbayes.github.io/tutorials/divrate/branch_specific.html, viewed on September 2021) (30). Detailed bioinformatic analyses and important assumptions are presented in the Supporting Information.

## Analysis of metagenome-assembled genomes

### Assembly, binning, and refinement

Quality-filtered sequences of 101 biofilms and 91 seawater metagenomes were assembled using metaWRAP v1.2, which integrates megahit and metaSPAdes, and further binned to MAGs using metaBAT2, CONCOCT, and MaxBin2 methods (31). Scaffolds ≥ 1,000 bp were retained for binning. MAGs with ≥50% completeness and ≤10% contamination scores were selected for refinement in metaWRAP. The refined medium- to high-quality MAGs were retained for downstream analyses (32). The process generated medium- to high-quality 77 marine biofilm-forming MAGs and 49 seawater MAGs.

### Taxonomic and morphological classification and functional annotation

Functional capacities of cyanobacteria lineages were analyzed across 77 marine biofilm-forming MAGs and 49 seawater MAGs, as well as 60 cyanobacterial genomes of a previously reported data set (33). The taxonomic classification of the MAGs was analyzed using GTDB-Tk v.1.5.0 (34). The functions of MAGs and reference genomes were annotated using the DRAM software with default parameters (35). PCoA was performed to identify overall functional dissimilarities between 77 marine biofilm-forming MAGs and 49 seawater MAGs with completeness >50%–90% was investigated, and completeness >75% was selected to display. Subsequently, a detailed investigation of the presence and absence of functional genes across these MAGs was conducted. For important functional genes yet to be integrated into a known database, we built a database (https://github.com/jchenek/ref-seqs-chromatic-acclimation) for pigment type identification with the marker gene cpcBA, as well as genes implicated in chromatic acclimation genetic islands, including mpeZ, mpeY, mpeW, and mpeQ (36). Then, the coding sequence of the MAGs was predicted using Prodigal v.2.6.3 by using the "-p meta"

parameter (37) and annotated with Diamond v.2.0.4 (38) by using the "--sensitive, -k 1, -e 1e-20, and 1e-60" parameters against the customized database, and genes with coverage of less than 75% were discarded.

## Phylogenetic analysis

To place the reconstructed biofilm-derived MAGs and seawater-derived MAGs into the phylogeny of cyanobacteria, we performed a phylogenetic analysis based on 27 widely used conservative genes for evolutionary analysis (39, 40). This method has higher accuracy of the phylogeny and timing of diversification compared to the phylogenetic analyses using only 16S rRNA genes. The reference sequences for these marker genes were downloaded from NCBI. The protein-coding genes of MAGs and reference genomes were predicted and extracted using Prodigal v2.6.3 (37). The extracted protein sequence of each coding gene was searched against the reference database of marker genes using PSI-BLAST with an e-value = $1e^{-05}$ and max_target_seqs = $1e^7$ to obtain the homologous genes in each genome (41). Subsequently, each set of homologous proteins was aligned using MAFFT v7.222 (42) and trimmed using trimAl v1.4 ((1, 43) using the auto options. A maximum likelihood (ML) phylogenetic tree was constructed using the IQ-TREE v2.1.2 (44). In addition, the best substitution model was searched and applied using the MFP + MERGE option, and a total of 1,000 ultrafast bootstrap replicates were sampled to evaluate the robustness of the phylogeny. The final results were visualized and rooted with iTOL v. 6.6 (45).

## Molecular dating

The divergence time of cyanobacteria was estimated with MCMCTree v. 4.9e (39) on 27 conservative genes previously proposed to be valuable to date bacterial divergence (46) and cyanobacteria phylogenetic trees (33). Since molecular dating analysis is known to be intrinsically associated with calibration points (47), and the calibrations of cyanobacteria, such as the fitness of different molecular clock models, have been tested in the previous study (33), we followed it to use the independent rate model for molecular clock analysis of cyanobacteria in this study. The Bayesian molecular clock analysis has been run twice for verification with a burn-in of 50,000 and a total of 500,000 generations. Based on the well-established methodology for molecular dating of cyanobacteria, we were able to select the most precise estimates of cyanobacteria evolutionary timeline for illustration and further discussion. The tree was visualized in FigTree v.1.4.4. All files and procedures used for the molecular dating analysis were uploaded to the github (https://github.com/ylifc/Biofim_cyanobacteria_evolution).

## Significant tests

For analyses associated with alpha-diversity and taxonomic profiling, ANOVA combined with TukeyHSD analysis was used to test whether the difference in diversity values observed between groups was significant. Similarly, ANOVA was used to test the significance of Principal Coordinate 1 (PC1) and Principal Coordinate 2 (PC2) used in PCoA. PERMANOVA was utilized to test the significance of PCoA results for the observed clusters. When PCoA was established, envfit correlation implemented in Vegan v. 2.5–7 in R was employed to search for relevance between the tested samples/genomes and separate factors. In these statistical tests, $P < 0.05$ was regarded as a significant difference.

## RESULTS

### Taxonomic compositions

The niche differentiation was revealed by a Venn diagram analysis, in which 346 operational taxonomic units (OTUs) were unique to marine biofilms, which was nearly ninefold higher than the number of OTUs specific to seawater samples (Fig. 1b). The

distribution of the full-length 16S rRNA gene units was similar to that of the OTUs. While the comparison showed that the relative abundance of cyanobacteria in 101 marine biofilms was considerably lower than that in 91 seawater, the relative abundance of cyanobacteria in rock substrata of the biofilms was threefold higher than those in seawater samples (Fig. 1c). Cyanobacteria in marine biofilms on rock surfaces had a tenfold higher fraction (31.4% of the total community average) than those on other substrata (Fig. S1).

Further taxonomic analyses showed that the cyanobacteria communities in seawater and biofilms displayed a profound difference in taxonomic compositions (Fig. 1d). A taxonomic difference based on 16S rRNA miTag was revealed at the subclass level profiling: Oscillatoriophycideae (39.9% of total community on average) and Nostocophycideae (1.29% of total community on average) were major lineages in biofilms, whereas Synechococcophycideae (95.1% of the total cyanobacteria on average) were dominant in seawater samples (for the difference at the genus level, see Fig. S2). On average, the relative abundance of the cyanobacteria (excluding Chloroplasts) and the taxonomic differentiation in biofilms (e.g., Oscillatoriales), as represented by 16S rRNA miTag, were in line with that using protein marker genes (Fig. S3). However, the differentiation depicted by the protein-coding marker was less than that resolved by 16S rRNA-based classification, and the stricter classification resulted in lower detection of cyanobacteria of sub-levels in some biofilm and seawater samples.

## Diversity analysis

The local diversity (i.e., alpha-diversity) of biofilm-forming cyanobacteria, including richness and diversity, was significantly ($P < 0.05$) lower than that of cyanobacteria in seawater samples. The Jaccard distance-based OTU composition dissimilarity, which represents the regional diversity of cyanobacteria in biofilms, was significantly higher ($P < 0.05$). Selected diversity indices are presented in Fig. 2a, and all measured values are presented in Table S1.

For beta-diversity analysis, the cyanobacterial community displayed a significant ($P < 0.001$) difference in principal coordinate analysis (PCoA) of the normalized OTU compositions for 10,000 sequences per sample (Fig. 2b). Results of further analyses show Synechococcophycideae and Oscillatoriophycideae species are key cyanobacterial lineages in separating marine biofilms and seawater samples on PCoA (summarized in Fig. S4).

## Generalists and specialists

In the OTU-based network analysis, the biofilm samples were clustered geographically, while seawater samples were tightly connected to each other regardless of their origins (Fig. S5, Table S2). The average betweenness centrality of biofilm samples (160) was twofold higher than that of seawater samples, but the average degree (32) and the eigenvector centrality (0.14) were lower than the indices (0.63 and 0.67) of seawater. These centrality properties for an individual are shown in Figs. S6 through S9.

The notion of less biogeographic connectivity of marine biofilm cyanobacteria relative to seawater counterparts was supported by Mantel tests with a distance of up to 15,000 km (Fig. S10). With comparable data sets, we observed a stronger correlation between geographic distance and cyanobacterial composition dissimilarities in the biofilm samples ($R^2 = 0.29$) than in the seawater samples ($R^2 = 010$). This demonstrated that the biofilm-forming cyanobacteria had a stronger distance decay than the seawater cyanobacteria.

With an adopted definition (see Materials and Methods), seawater contained both specialists and generalists, while biofilms only contained specialists, and the specialists in biofilms generally had a higher abundance than those in seawater (Fig. 3a and b). As also evident in Fig. S11, the steeper decreasing trend of the same cyanobacteria OTU in increasing marine biofilms ($R^2 = 0.70$) than seawater ($R^2 = 0.95$) suggests that the cyanobacteria OTUs in seawater were simultaneously present in more samples than the

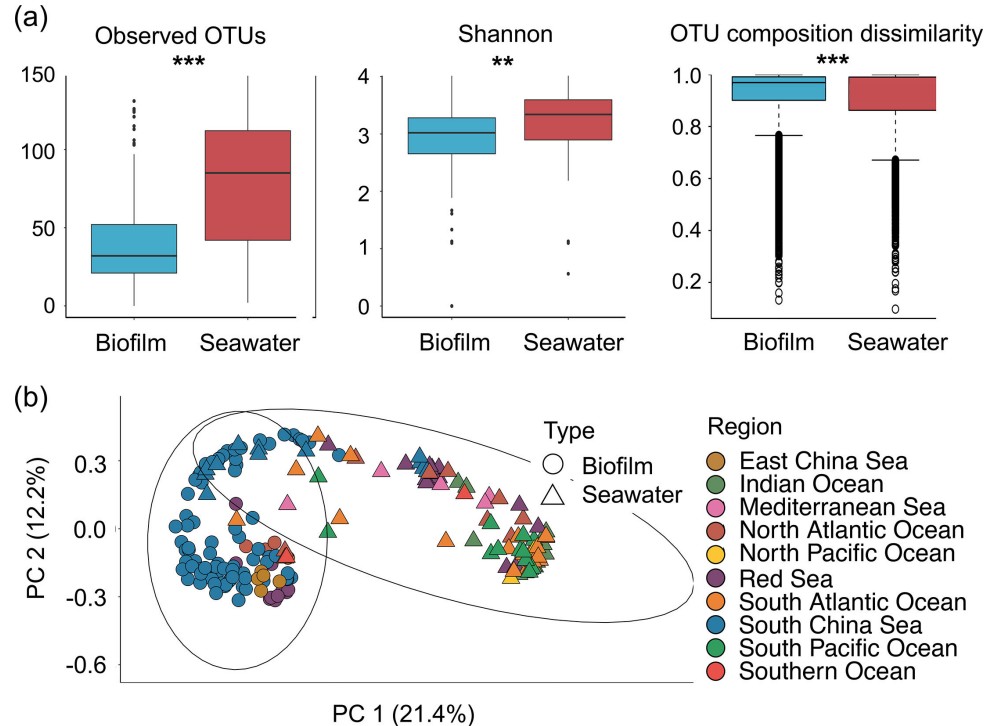

FIG 2 Diversity analysis between marine biofilm and seawater samples. (a) Bray–Curtis distance of the cyanobacterial communities shown by principal coordinate analysis (PCoA) of the normalized OTU matrix; 95% confidence intervals for the biofilm and seawater samples are present in the PCoA ordination. (b) Comparison of observed OTUs, Shannon diversity, and Jaccard distance-based OTU dissimilarity.

biofilm OTU (i.e., the same cyanobacteria OTU is present in more seawater samples than it could be present in the biofilm sample). More detailed analyses showed that the relative abundance of cyanobacteria is positively correlated to biofilm development duration, and a high abundance of cyanobacteria was found in the samples collected in spring and summer (Fig. S12). Biofilm-forming cyanobacteria in the Red Sea displayed a more diverse and niche partitioning pattern than seawater counterparts (Fig. S13, also see the detailed results and discussion in the supplemental material).

## MAG characteristics and functional analysis

In total, we generated 1,047 marine biofilm MAGs and 1,601 seawater MAGs of medium–high quality (32); among these MAGs, 77 biofilms and 49 seawater MAGs were classified as cyanobacteria. This result substantially increased the total number of cyanobacteria MAGs derived from marine biofilms by eight times compared to a previous study, which used the same sequences to illustrate the overall biodiversity in marine biofilms (15). The detailed properties of newly generated MAGs are presented in the supplemental material. A broader comparison of the MAG sequences also showed substantial differences in taxonomy (Fig. 4a) and functional potential (Fig. 4b) of cyanobacteria across marine biofilms and seawater. Sequences of biofilm-forming cyanobacteria were assigned to diverse taxonomy, while seawater-derived cyanobacteria were mostly assigned to *Synechococcus* and *Prochlorococcus*. Based on the phylogenomic tree of cyanobacteria been referred from two sets of reference genomes (6, 33), the biofilm-forming MAGs covered the phylogenetic diversity of different evolutionary stages, especially the deep branches of this phylum, whereas seawater-derived MAGs mainly formed clusters in shallow branches. Furthermore, with the support of the bootstrap values, these phylogenetic trees consistently demonstrate the novelty of phylogeny of the cyanobacterial lineages in biofilm-derived MAGs (Fig. 4c; Fig. S14).

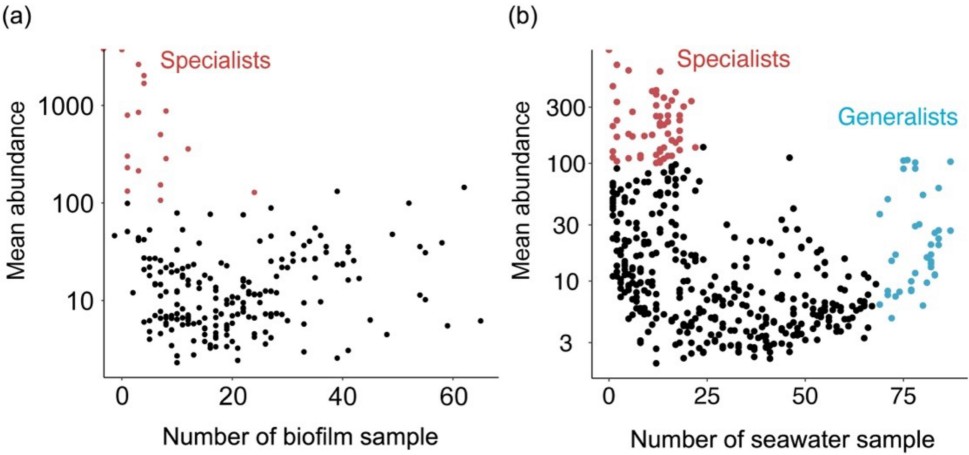

**FIG 3** Occupancy patterns of the cyanobacterial communities across (a) biofilm and (b) seawater samples. Abundance (y-axis) and occupancy (x-axis) plots for the 16S rRNA miTags OTU. Habitat generalist OTUs (in blue) are defined in >75% of the total sample size, and specialist OTUs (in red) are defined in <25% of the total sample size with a relative abundance of >100 in log 10 scales.

The difference in functional gene contents between 43/77 biofilm-derived and 13/49 seawater-derived cyanobacteria was presented on the PCoA for MAGs with completeness >75% (Fig. 4d). The composition of functional genes of the biofilm MAGs with decreasing completeness tended to be similar to the composition of the seawater MAGs, which is consistent with the observation of substantial reductions in genome size and functional richness from biofilm MAGs to seawater counterparts (Fig. 4d). Completeness, contamination, and GC content of all MAGs are presented in Fig. S15. The major difference in the functional capacity between biofilm MAGs and seawater MAGs was that the biofilm MAGs generally had more complete functional genes related to Fe and Zn compared to seawater MAGs (Fig. 4d). The difference of other transport genes for oxygen-sensitive elements such as Ni and Mn (for Ni and Mn transport gene annotation, see Fig. S15), or nutrients such as P, between biofilm-forming and seawater MAGs was not observed. With regards to chromatic acclimation (light adaptation), seawater MAGs housed both pigment type 3 c and pigment type 3d genes, whereas pigment genes were not detected in any biofilm MAGs (Fig. 4d).

## Molecular dating, diversification rate, and links with geological events

Bayesian molecular dated phylogenetic trees by using MAGs (Fig. 5a, see Fig. S16 for a full version tree) and full-length 16S rRNA genes (Fig. 5b, see Figs. S17 and S18 for full version trees) showed that marine biofilm-forming cyanobacteria evolution may be coincident with major changes in Earth's past climate. The OTU prevalence (Fig. 5b) is consistent with the distribution of MAG-based taxonomy (Fig. 4a), suggesting that a few genera of close phylogeny dominate seawater cyanobacteria. The age estimates at critical branching points were consistent between the two trees, but the MAG-based tree provided a higher phylogenetic resolution to identify previously unrecognized cyanobacteria diversification events.

Three critical time points, 2.5 Ga, 1.3 Ga, and 0.8 Ga, were identified for marine cyanobacteria evolution. The former coincides with the Great Oxidation Event (GOE) when Earth's atmosphere first became oxygenated (48). Cyanobacteria derived from marine biofilm MAG show them to be the first lineage among all studied genomes after the occurrence of GOE (Fig. 5a). This was followed at 1.3 Ga by a significant diversification of biofilm cyanobacterial lineages (Fig. 5a). Despite that, there is an interesting missing diversification activity of biofilm-forming cyanobacteria between GOE and 1.3 Ga. This is surprising given that this period of time includes two major oxygen-related events, the "overshot of oxygen" associated with the Lomagundi Event between 2.2 and 2.06 Ga

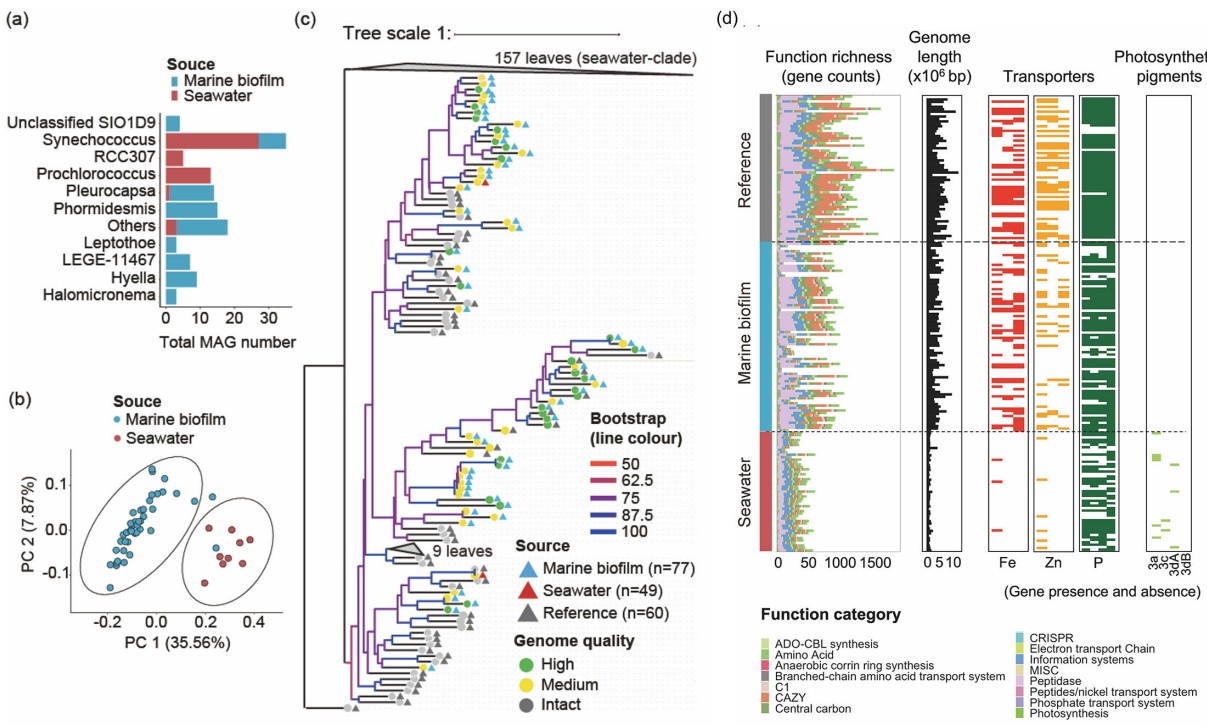

**FIG 4** The novelty, characteristics, and functional analyses of 126 generated metagenome-assembled genomes (MAGs). The overall distribution of (a) GTDB-Tk (34) taxonomic assignments (total MAG numbers < 3 is defined as "others") and (b) functional genes for 77 cyanobacterial MAGs from marine biofilms and 49 cyanobacterial MAGs from seawater. The PCoA ordination is made by Bray–Curtis distance for biofilm and seawater MAGs > 75% completeness based on the sum of functional genes classified into carbon utilization, transporters, energy, organic nitrogen, and miscellaneous functional categories by using the DRAM software (35). (c) Phylogenetic tree of MAGs across marine biofilms and seawater with bootstrap values and morphology characterization. Branches with bootstrap values < 50 were collapsed. (d) MAG size and functional gene comparison with a focus on biological essential element transporters and photosynthesis adaptation across all MAGs and reference genomes. Functions with the top 15 abundant genes are retained for functional profiling overview.

where atmospheric $O_2$ levels may have increased above 10% of present atmospheric levels, and then there is a drop in atmospheric $O_2$ at 2.0 Ga (49). The third critical point marks the latest large diversification of marine biofilm cyanobacteria that occurred soon after the Neoproterozoic Oxidation Event (NOE) at about 0.8 Ga, which was in coincidence with the emergence and fast diversification of the seawater-derived MAGs (Fig. 5a). By estimating the lineage-specific diversification rates, we observed a significant increase in the speciation rate at the largest subclade of the seawater-derived cyanobacteria, which is coincident with the NOE (Fig. 5b).

## DISCUSSION

### Cyanobacterial biodiversity

We demonstrate that marine biofilms act as critical cyanobacteria factories in global oceans, underpinning complex biosphere and marine environments throughout history. This study revealed an overall picture of metacoupling of the evolutionary, functional, and ecological patterns of marine cyanobacteria and critical geological events (Fig. 6). Compared with the analysis of isolated genomes, metagenomic analysis has made it possible to examine unculturable genomes and further link the genomes of individuals/ groups to specific environments in an ecological context (50, 51). While exploration of marine biofilms has dramatically increased the known microbial species (15), with improved bioinformatic processing methods, this study further promotes these insights as represented by high-quality MAGs of cyanobacteria. The Venn diagram also supports that marine biofilms may hold a large and unique, underappreciated cyanobacterial

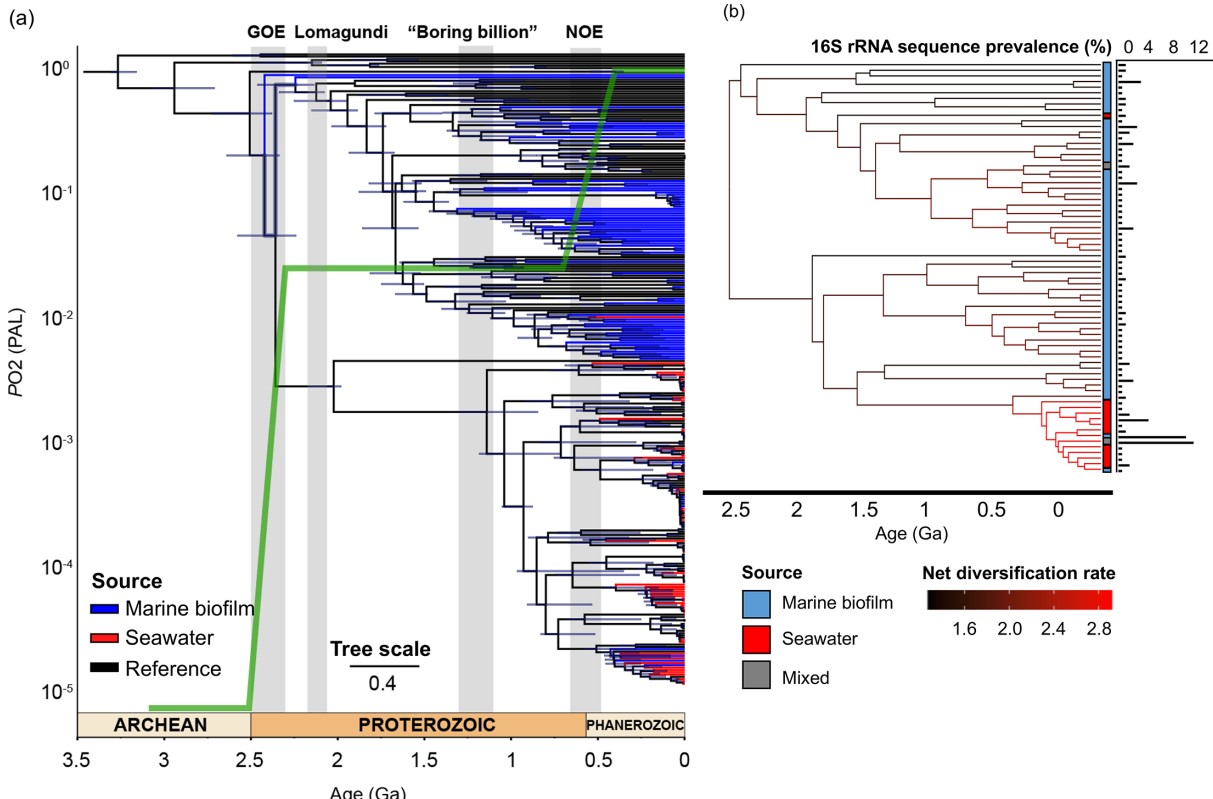

**FIG 5** Bayesian molecular dated phylogenetic tree of cyanobacteria across marine biofilm and seawater samples. (a) Molecular dated phylogenetic tree of 77 MAGs derived from 101 marine biofilms and 49 MAGs derived from 91 seawater samples from global oceans with 60 references of full genome sequences (33). The MAG source is represented by the colored branches, and the diversification intensity for cyanobacteria from different sources is indicated by the color density. (b) Full-length 16S rRNA gene-based molecular dated phylogenetic tree of 62 sequences extracted from metagenomes of biofilm samples and 13 sequences from metagenomes of seawater samples with 56 references of 16S rRNA gene sequences (4); an estimation of changes in net diversification rates in major tree branches was done. Geological events associated with stepwise oxygenation (the green line), as represented by atmospheric oxygen (PAL), and their estimated occurrence time refer to previous studies (48).

abundance and diversity for coastal regions and estuaries compared to seawater. The implementation of full-length 16S rRNA genes (100% similarity) in the Venn diagram has mitigated the bias associated with planktonic cyanobacteria with highly similar genomes, such as *Prochlorococcus* data. Compared with other forms of benthic life, marine biofilm development is highly dynamic (52). The cyanobacteria functioning and their carbon export activity in the future ocean may be altered in the contemporary ocean ecosystem that undergoes rapid changes in trace metal contents due to the growing human population and industrialization (53). Thus, the results indicate that the unknown cyanobacterial population remains large, and it is critically important for re-assessments of primary production and other biogeochemical budgets of the oceans (54, 55).

## Cyanobacterial adaptation represented by oceanic niche differentiation

Cyanobacterial adaption strategies are diverse, as previously recorded by *Synechococcus* and *Prochlorococcus* (56–58). The newly explored niche differentiation of cyanobacteria contributes to further understanding of cyanobacterial adaptation for global oceans. Analyses indicate more geographical constraints imposed on biofilm-forming cyanobacteria than their seawater counterparts and biofilm-forming lineages were less dispersed in the oceans. Generalists may promote the evolution of a microbial community (21), and the diversification rate estimation (Fig. 5b) fits well with this recent generalist–specialist evolution model, suggesting that marine biofilm-derived cyanobacteria may optimally

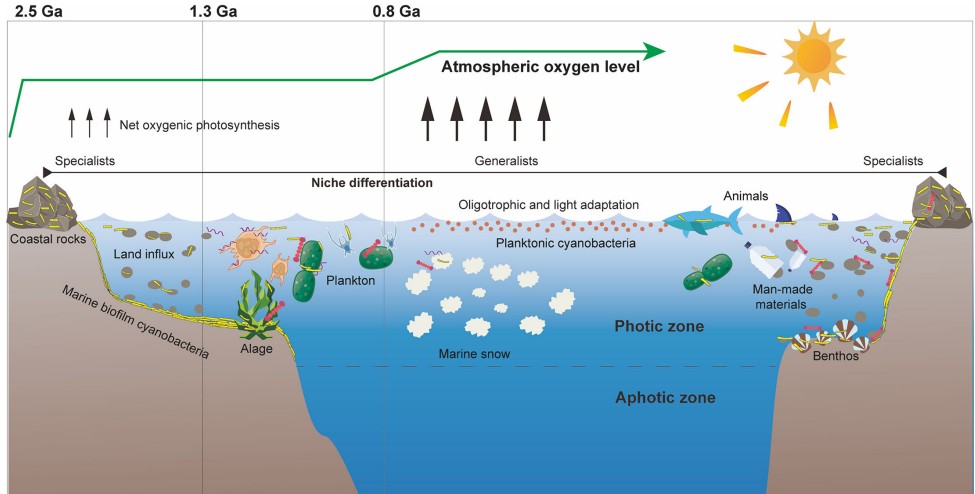

**FIG 6** A concept model of the marine biofilms as cyanobacterial factories in the marine environment. This figure elucidates marine cyanobacteria ecology—evolution nexus across oceanic niches, spatial and geological time scales on the basis of the metagenomic data analysis, highlighting the seeds and diversification of cyanobacteria to the oceans through marine biofilms at critical time intervals at 2.5 Ga, 1.3 Ga, and 0.8 Ga, and various vectors (e.g., coastal rocks, suspended particles, animal surface, and seafloor) that may provide a surface for marine biofilms for shielding large cyanobacterial biomass in the ancient and modern oceans.

function in geographically constrained regions; in contrast, high diversification rates of seawater-derived cyanobacteria have a significant role in facilitating cell dispersion across environmental barriers (59). The observed reduced functional richness and genome size align with the reductive genome evolution phenomena (genomic stream-lining) prevalent in marine bacterioplankton (60), and for marine planktonic cyanobacteria specifically, with new characteristics such as decreased cell diameter and loss of filamentous forms, and then evolved and spread across the oceans (61, 62). Our results show that the most significant differences in functional potential are related to various transporters (e.g., Fe), which may be caused by the uneven distribution of nutrients for global oceans and/or continuous changes in oceanic geochemistry throughout Earth's history. These ecological and functional potential patterns infer that "geo-bio" co-evolution may significantly determine the development of the biofilm-forming cyano-bacteria.

## Cyanobacteria evolution pattern in marine biofilms

Our results suggest that the newly explored cyanobacterial diversity is closely related to the evolution of cyanobacteria throughout Earth's history, and the reconstructed picture clearly displays the complexity of the marine cyanobacteria evolution. The rock record shows a substantial rise of planetary oxygenation toward the end of Archean, the so-called Great Oxidation Event, or GOE. It is widely accepted that a substantial population of planktonic cyanobacteria in the Archean water column was already required to generate that amount of oxygen (7, 63). Based on our result, the seeding of early planktonic cyanobacteria may have come from marine biofilms around the GOE, either growing on the seafloor in near-shore coastal settings and/or attached to increased flux of oceanic particles in seawater (3, 64, 65). Ancient cyanobacterial genomes older than 2.5 Ga were not found (for more discussion on geochemical aspects, see the supplemental material) (66).

The evolutionary trajectory of cyanobacteria challenges the view of slow biological evolution in the Earth's middle Proterozoic between 1.8 Ga and 0.8 Ga, that is, the so-called "boring billion", since there was a major diversification at 1.3 Ga (Fig. 5A). Our data suggest that the second remarkable diversification of marine biofilm-derived

cyanobacteria was followed by the emergence of eukaryotes that inhabited the world's oceans as early as 1.8–1.6 Ga, which was recorded by fossil records (7) and likely to be driven by the endosymbiotic event established ~1.9–1.25 Ga (63, 67). From a biological perspective, forming biofilms may have significant advantages in establishing a symbiotic relationship and other interactions with eukaryotes(52).

The long lag for planktonic cyanobacteria to evolve from marine biofilms/associated symbionts may essentially link to declining metal availability in Mesozoic seawater (8, 68, 69), as evidenced by the functional divergence herein, such as the minimized requirement for Fe and Zn transporters in seawater cyanobacteria (Fig. 4D) and the minimized proteomic requirement of these trace metals recorded elsewhere (55). This is also evidenced by the wide detection of cyanobacterial lineages located in the early branching points on the evolutionary tree in the Red Sea samples, as these samples are derived from upper water columns of the brine pools with uniquely high concentrations of metals (70–72). Cyanobacteria evolution is multidirectional in the modern oceans, as our data also suggest a previously overlooked and fast evolution event of marine biofilm-derived cyanobacteria after NOE.

## Limitations and future steps

Our results highlight the importance of continuously exploring the unknown cyanobacterial population in marine biofilms and other underexplored niches in the marine environment. Geographical and substrata unevenness of sampling may cause bias in the results. For example, most of the biofilms are from SCS, and the seawater samples from SCS are much closer to the SCS biofilm samples than the others. More marine biofilm sampling worldwide is required. While we focused on the overall picture of the cyanobacteria in the marine biofilms, many details on the functional potential and bio-geo interactions are yet to be explicated. Also note that the molecular-dated trees adopted numerous methods previously established, and in fact, many areas, such as the key timing of cyanobacterial diversification, remain controversial. As for interesting patterns revealed in our results, implementing alternative clock models is the next step to constrain the important evolution timing. Nevertheless, given the prevalence of marine biofilms, this study shall serve as a foundation for further investigation of the critical roles of cyanobacteria and other key microbial participants in Earth's changing oceans and climates.

## ACKNOWLEDGMENTS

We thank Dr. R. Zhang and Dr. M. Matsui, who provided technical assistance and invaluable comments on the manuscript.

This research was supported by the Major Project of Basic and Applied Basic Research of Guangdong Province (2019B030302004), Southern Marine Science and Engineering Guangdong Laboratory (Guangzhou) 880 (2021HJ01, SMSEGL20SC01) awarded to P.-Y.Q. W.I. was supported by Japan Science and Technology Agency (JPMJCR19S2) and Japan Society for the Promotion of Science (19H05688 and 18H04136). The work described in this paper was supported by a fellowship award from the Research Grants Council of the Hong Kong Special Administrative Region, China (HKUST PDFS2223-6S03), awarded to C.Z.

C.Z. wrote the paper with input from all authors. S.Y. performed 16S rRNA-based phylogenetic analyses, and Y.L. performed MAG-based phylogenetic analyses. J.C. performed cell counting, A.C. performed SEM imaging, and T.W., R.W., and K.Z. performed data processing. S.Y., W.H., H.L., W.I., K.O.K., and P.-Y.Q. edited the paper. C.Z. and P.-Y.Q. designed the study.

## AUTHOR AFFILIATIONS

[1]Department of Ocean Science, The Hong Kong University of Science and Technology, Hong Kong, China

²Southern Marine Science and Engineering Guangdong Laboratory, Guangzhou, China

³Department of Biological Sciences, Graduate School of Science, The University of Tokyo, Bunkyo-ku, Tokyo, Japan

⁴Department of Earth and Atmospheric Sciences, Faculty of Science, University of Alberta, Edmonton, Alberta, Canada

⁵Department of Integrated Biosciences, Graduate School of Frontier Sciences, The University of Tokyo, Kashiwa, Chiba, Japan

## AUTHOR ORCIDs

Cheng Zhong  http://orcid.org/0000-0002-8709-7444
Hongbin Liu  https://orcid.org/0000-0002-3184-2898
Pei-Yuan Qian  http://orcid.org/0000-0003-4074-9078

## FUNDING

| Funder | Grant(s) | Author(s) |
| --- | --- | --- |
| Southern Marine Science and Engineering Guangdong Laboratory (Guangzhou) (南方海洋科学与工程广东省实验室) | 2021HJ01, SMSEGL20SC01 | Pei-Yuan Qian |
| MEXT \| Japan Science and Technology Agency (JST) | JPMJCR19S2 | Wataru Iwasaki |
| MEXT \| Japan Society for the Promotion of Science (JSPS) | 19H05688, 18H04136 | Wataru Iwasaki |
| Research Grants Council, University Grants Committee (研究資助局) | HKUST PDFS2223-6S03 | Cheng Zhong |

## AUTHOR CONTRIBUTIONS

Cheng Zhong, Conceptualization, Data curation, Formal analysis, Funding acquisition, Investigation, Methodology, Validation, Visualization, Writing – original draft, Writing – review and editing.

## DATA AVAILABILITY

101 biofilm metagenomes and 24 adjacent seawater metagenomes refer to the publicly available data sets in the NCBI database under BioProject accession no. PRJNA438384 (BioSample no. SAMN08714533 for the 101 biofilm metagenomes; BioSample no. SAMN08714535 and SAMN08714533 for the 24 adjacent seawater metagenomes); 67 Tara Oceans seawater samples refer to the publicly available data sets in the EBI under the project identifiers PRJEB402 and PRJEB7988. Additional data are available in Figshare (https://doi.org/10.6084/m9.figshare.21485238.v2) as follows. The metadata of the metagenomes re-analyzed in this study are presented in Supplementary Data A. The data of 16S rRNA miTags taxonomic, diversity, and ecological analyses are provided in Supplementary Data B-D. The data and accession number of 77 marine biofilm-derived cyanobacterial MAGs and 49 seawater-derived cyanobacterial MAGs are provided, and their detailed properties are given in Supplementary Data E. The accession numbers of each of the reference sequences and genomes used for the 16S rRNA-based evolutionary analyses and the MAG-based evolutionary analyses are listed in Supplementary Data F. A full result of functional gene annotation of MAGs is listed in Supplementary Data G.

## ADDITIONAL FILES

The following material is available online.

## Supplemental Material

**Supporting information (mSystems00317-24-S0001.docx).** Fig. S1-S18, Tables S1 and S2, and additional experimental details.

## Open Peer Review

**PEER REVIEW HISTORY (review-history.pdf).** An accounting of the reviewer comments and feedback.

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
