## [Reviewer comments · mSystems]

Marine biofilms: cyanobacteria factories for the global oceans

Cheng Zhong, Yamanouchi Shun, Yingdong Li, Jiawei CHEN, Tong Wei, Ruojun Wang, Kun Zhou, Aifang Cheng, Weiduo Hao, Hongbin Liu, Kurt Konhauser, Wataru Iwasaki, and Pei-Yuan Qian

Corresponding Author(s): Cheng Zhong, Southwest Petroleum University

Review Timeline:

Submission Date:	March 25, 2024
Editorial Decision:	May 24, 2024
Revision Received:	August 14, 2024
Accepted:	September 6, 2024

Editor: Michaeline Albright

Reviewer(s): The reviewers have opted to remain anonymous.

Transaction Report:

DOI: <https://doi.org/10.1128/msystems.00317-24>

Re: mSystems00317-24 (Marine biofilms: cyanobacteria factories for the global oceans)

Dear Dr. Zhong cheng:

Thank you for the privilege of reviewing your work. Below you will find my comments, instructions from the mSystems editorial office, and the reviewer comments. Overall, the data analysis/results were compelling. However, for publication the introduction must be significantly improved in order to provide a clear justification and framing of the scientific purpose of the work.

Revision Guidelines

Sincerely,
Michaeline Albright
Editor
mSystems

Reviewer #1 (Comments for the Author):

Major Comments:

In the submitted article, authors Zhong et al. examine metagenomic 16S rRNA miTags and MAGs in order to compare compositional diversity and functional potential between surface-associated marine cyanobacterial biofilms and free-living cyanobacteria in seawater. They also use their MAGs to construct a cyanobacterial phylogenetic tree and apply molecular

dating to determine dates of diversification and divergence. Overall, the authors apply cutting edge bioinformatic techniques in line with the standards of the field. Their analyses are robust and the figures displaying their results are truly exceptional. Some of their results are perhaps unsurprising, including that free-living cyanobacteria are more diverse and that biofilm-associated cyanobacteria have greater endemism, dispersal limitation, and beta-diversity. However, these are still important results to report and set the stage for their broader discussion at the end of the paper.

The most exciting components of the article are figures 4-6, which demonstrate deep divergences in the cyanobacterial phylogenetic tree that are associated with major shifts in not only genome length and functional potential, but also major geologic events in Earth's history. This work is transformative in that it not only shows diversification aligned with expected events such as the Great Oxidation Event and the Neoprotozoic Oxidation Event, but also during unexpected time periods such as the "boring billion." This unexpected diversification is worth further study and may be an interesting avenue of research in the future.

Despite the strengths in the analysis and results of the study, the article's greatest weakness is in the foundational theoretical framing of the work. Although the authors spend a great deal of their results and discussion examining the evolutionary history of cyanobacterial lineages, the importance of cyanobacteria for geologic history is barely addressed in the introduction. In fact, after reading the introduction, I would struggle to tell you why the authors decided to perform this study beyond the fact that marine cyanobacterial biofilms "are previously underappreciated niches." I do not find this argument to be a particularly compelling reason to justify a scientific study. I would suggest that the authors completely re-write their introduction to clearly identify the fundamental eco-evolutionary questions that they wished to address with this study. I would also expect their introduction to spend more time placing their work in the context of previous studies that have examined the geologic history of cyanobacterial lineages across Earth's history.

Minor Comments:

Line 44: "which is in stark..."

Line 68: "functional potential."

Line 70-76: Numerous grammatical errors in these two sentences, please revise. I also more commonly see "surface-associated microbes" rather than "surface association microbes"

Line 79: "the Archean"

Lines 82 and 86: "composition"

Line 87: "evolutionary"

Lines 86-87: The "ecological and evolutionary" implications of what exactly? The existence of surface-associated cyanobacteria? Please elaborate further as this is a vague statement and the goals of this study are unclear.

Line 171 (and throughout): "seawater" or "seawater samples" (not "seawaters")

Line 219: "lineage"

Line 324: "Chloroplasts"

Lines 335 and 348: "indices"

Line 337: "a significant"

Line 348: 0.63?

Line 373: The second half of this sentence seems to be missing a verb? How are the 77 and 49 MAGs different from the other MAGs?

Line 389: "gene content"

Line 394: "GC content"

Line 466: Genomic streamlining is a more appropriate concept to cite here rather than "small is beautiful"

Line 469: "functional potential"

Line 502: What is an "accident cyanobacterial lineage"?

Lines 504-506: This sentence is unclear and poorly supported, either move entirely to supplemental information or explain further in the main text.

Reviewer #2 (Comments for the Author):

The authors have comprehensively re-analyzed a cyanobacterial metagenomic dataset from marine biofilm and planktonic samples, extracting information on diversity, specialists vs generalists (niche differentiation) and the timing of diversification across geological deep time. The last part especially is rather speculative, but that's the nature of the field. The authors explain their assumptions and I think they make all the appropriate disclaimers. I found this well-written, interesting and thought-provoking. I'm not a specialist in the data analysis methods used, so I don't have any technical corrections to suggest. My only issue is that I think the last sentence of the Importance statement "surface associated biomass within oceanic niches emerges as a pivotal factor in Earth's evolution" goes too far. If the authors' interpretation is correct and there are indeed bursts of diversification that coincide with major biogeochemical transitions, that doesn't necessarily mean that cyanobacterial diversification in marine biofilms was driving those transitions, as the authors seem to be implying. It could just be that the evolution of biofilm cyanobacteria was accelerated by changes in the biosphere induced by other factors.

Manuscript ID mSystems00317-24

Response to Reviewers

Dear Dr. Michaeline Albright

Editor

mSystems

We would like to submit a revised manuscript, "**Marine biofilms: cyanobacteria factories for the global oceans**", for publication in the mSystems. We appreciate the time and effort that you and the reviewers dedicated to providing insightful comments to improve our manuscript. We have addressed all the comments raised by the reviewers, and those changes are highlighted in blue texts. Please see below, in blue, for a point-by-point response to the reviewers' comments.

Reviewer 1

Comments to the Author

In the submitted article, authors Zhong et al. examine metagenomic 16S rRNA miTags and MAGs in order to compare compositional diversity and functional potential between surface-associated marine cyanobacterial biofilms and free-living cyanobacteria in seawater. They also use their MAGs to construct a cyanobacterial phylogenetic tree and apply molecular dating to determine dates of diversification and divergence. Overall, the authors apply cutting edge bioinformatic techniques in line with the standards of the field. Their analyses are robust and the figures displaying their results are truly exceptional. Some of their results are perhaps unsurprising, including that free-living cyanobacteria are more diverse and that biofilm-associated cyanobacteria have greater endemism, dispersal limitation, and beta-diversity. However, these are still important results to report and set the stage for their broader discussion at the end of the paper.

The most exciting components of the article are figures 4-6, which demonstrate deep divergences in the cyanobacterial phylogenetic tree that are associated with major shifts in not only genome length and

functional potential, but also major geologic events in Earth's history. This work is transformative in that it not only shows diversification aligned with expected events such as the Great Oxidation Event and the Neoprotozoic Oxidation Event, but also during unexpected time periods such as the "boring billion." This unexpected diversification is worth further study and may be an interesting avenue of research in the future.

Despite the strengths in the analysis and results of the study, the article's greatest weakness is in the foundational theoretical framing of the work. Although the authors spend a great deal of their results and discussion examining the evolutionary history of cyanobacterial lineages, the importance of cyanobacteria for geologic history is barely addressed in the introduction. In fact, after reading the introduction, I would struggle to tell you why the authors decided to perform this study beyond the fact that marine cyanobacterial biofilms "are previously underappreciated niches." I do not find this argument to be a particularly compelling reason to justify a scientific study. I would suggest that the authors completely re-write their introduction to clearly identify the fundamental eco-evolutionary questions that they wished to address with this study. I would also expect their introduction to spend more time placing their work in the context of previous studies that have examined the geologic history of cyanobacterial lineages across Earth's history.

Author response: We thank you for sharing your positive feedback and valuable comments. In the revised manuscript, we substantially revised the introduction of the manuscript to identify the fundamental eco-evolutionary questions we addressed in this study (Lines 51-90). Particularly, we revised the introduction to place our work in the context of previous studies that have examined the geologic history of cyanobacterial lineages across Earth's history. The detailed revisions are shown in our point-by-point responses.

1. Line 44: "which is in stark..."

Author response: Thank you for your valuable feedback on our grammar. We have addressed the suggested corrections. Please note that, in response to comments 1-7, we have significantly revised the Introduction section as recommended by the reviewer, which may affect the applicability of some of the earlier grammar comments.

2. The Line 68: "functional potential."

Author response: Thank you we have revised the word to “functional potential” throughout the manuscript.

3. Line 70-76: Numerous grammatical errors in these two sentences, please revise. I also more commonly see "surface-associated microbes" rather than "surface association microbes"

Author response: Thanks for this comment. We have revised the grammar errors in these two sentences, including changing “surface association microbes” to “surface-associated microbes”. (Lines 67-80)

4. Line 79: "the Archean"

Author response: We revised the word to “the Archean” as suggested.

5. Lines 82 and 86: "composition"

Author response: we have revised the word to “composition” as suggested.

6. Line 87: "evolutionary"

Author response: we have revised the word to “evolutionary” as suggested.

7. Lines 86-87: The "ecological and evolutionary" implications of what exactly? The existence of surface-associated cyanobacteria? Please elaborate further as this is a vague statement and the goals of this study are unclear.

Author response: Thanks for the comment. The “ecological and evolutionary patterns of the surface-associated cyanobacteria. We have revised the sentence to clarify the goal of this study.

8. Line 171 (and throughout): "seawater" or "seawater samples" (not "seawaters")

Author response: We have used “seawater samples” or “seawater” throughout the manuscript.

9. Line 219: "lineage"

Author response: we have revised the word as suggested. (Line 215)

10. Line 324: "Chloroplasts"

Author response: we have revised the word as suggested. (Line 317)

11. Lines 335 and 348: "indices"

Author response: We have revised “indexes” to “indices” throughout the manuscript. (Lines 327 and 341)

12. Line 337: "a significant"

Author response: we have revised the word to “a significant”. (Line 330)

13. Line 348: 0.63?

Author response: Thanks for raising the issue, we corrected “63” to “0.63”. (Line 341)

14. Line 373: The second half of this sentence seems to be missing a verb? How are the 77 and 49 MAGs different from the other MAGs?

Author response: we revised the sentence for clarification. In total, we generated 047 marine biofilm MAGs and 1601 seawater MAGs. Among these MAGs, 77 biofilms and 49 seawater MAGs were classified as cyanobacteria. (Lines 365-367)

15. Line 389: "gene content"

Author response: we revised the word accordingly. (Line 382)

16. Line 394" "GC content"

Author response: we revised the word accordingly. (Line 387)

17. Line 466: Genomic streamlining is a more appropriate concept to cite here rather than "small is beautiful"

Author response: Thanks for the suggestion. We agree and revise this part accordingly. (Lines 455)

18. Line 469: "functional potential"

Author response: we have revised the word to "functional potential patterns" (Line 461)

19. Line 502: What is an "accident cyanobacterial lineage"?

Author response: we have rephrased the words to "cyanobacterial lineages located in the early branching points on the evolutionary tree" for clarification. (Line 491-492)

20. Lines 504-506: This sentence is unclear and poorly supported, either move entirely to supplemental information or explain further in the main text.

Author response: Thank you for the feedback. We agree with the reviewer that the statement was lacking context. We have moved the statement entirely in the Supplementary Information of the revised manuscript.

Reviewer 2

Comments to the Author

The authors have comprehensively re-analyzed a cyanobacterial metagenomic dataset from marine biofilm and planktonic samples, extracting information on diversity, specialists vs generalists (niche differentiation) and the timing of diversification across geological deep time. The last part especially is rather speculative, but that's the nature of the field. The authors explain their assumptions and I think they make all the appropriate disclaimers. I found this well-written, interesting and thought-provoking. I'm not a specialist in the data analysis methods used, so I don't have any technical corrections to suggest. My only issue is that I think the last sentence of the Importance statement "surface associated biomass within oceanic niches emerges as a pivotal factor in Earth's evolution" goes too far. If the authors' interpretation is correct and there are indeed bursts of diversification that coincide with major biogeochemical transitions, that doesn't necessarily mean that cyanobacterial diversification in marine biofilms was driving those transitions, as the authors seem to be implying. It could just be that the evolution of biofilm cyanobacteria was accelerated by changes in the biosphere induced by other factors.

Author response: Thank you for providing your insightful feedback. We agree with the reviewer that the last sentence of the importance statement may go too far. For clarification, we removed the last sentence of the importance statement "surface associated biomass within oceanic niches emerges as a pivotal factor in Earth's evolution" in the revised manuscript.

We hope these revisions address the reviewers' comments adequately and improve the quality of our manuscript. Thank you again for your consideration.

Sincerely,

Cheng Zhong and Co-authors

Re: mSystems00317-24R1 (Marine biofilms: cyanobacteria factories for the global oceans)

Dear Dr. Cheng Zhong:

Your manuscript has been accepted, and I am forwarding it to the ASM production staff for publication. Your paper will first be checked to make sure all elements meet the technical requirements. ASM staff will contact you if anything needs to be revised before copyediting and production can begin. Otherwise, you will be notified when your proofs are ready to be viewed. The reviewers have provided several grammatical/style suggestions that we would appreciate that you consider as you are correcting your proofs.

Sincerely,
Michaeline Albright
Editor
mSystems

Reviewer #1 (Comments for the Author):

I thank the co-authors for their timely and thorough responses to my previous review. The Introduction is improved from the previously submitted version. Given the newly revised Introduction, I have some minor grammatical and style suggestions for this section:

Line 51: "ecology and evolution"

Line 61: "oceanic chemistry and the biosphere"

Line 63: "However, understanding"

Line 71: delete "garnered"

Line 75-76: "Marine biofilms offer microbes numerous"...

Reviewer #2 (Comments for the Author):

I'm happy with the revisions. It's a very interesting and well-illustrated article. A couple of minor corrections I spotted as I was reading through it:

1. Line 70-71: "surface-associated microbes have garnered increasingly been recognized for" should be either "have garnered increasing recognition for" or "have increasingly been recognized for"
2. Fig. 6 - mis-spelling of "algae"